# Benzimidazole-Triazole Hybrids as Antimicrobial and Antiviral Agents: A Systematic Review

**DOI:** 10.3390/antibiotics12071220

**Published:** 2023-07-22

**Authors:** Maria Marinescu

**Affiliations:** Department of Organic Chemistry, Biochemistry and Catalysis, Faculty of Chemistry, University of Bucharest, 030018 Bucharest, Romania; maria.marinescu@chimie.unibuc.ro

**Keywords:** benzimidazole, triazole, hybrids, antimicrobial, antiviral, pharmaceutical properties

## Abstract

Bacterial infections have attracted the attention of researchers in recent decades, especially due to the special problems they have faced, such as their increasing diversity and resistance to antibiotic treatment. The emergence and development of the SARS-CoV-2 infection stimulated even more research to find new structures with antimicrobial and antiviral properties. Among the heterocyclic compounds with remarkable therapeutic properties, benzimidazoles, and triazoles stand out, possessing antimicrobial, antiviral, antitumor, anti-Alzheimer, anti-inflammatory, analgesic, antidiabetic, or anti-ulcer activities. In addition, the literature of the last decade reports benzimidazole-triazole hybrids with improved biological properties compared to the properties of simple mono-heterocyclic compounds. This review aims to provide an update on the synthesis methods of these hybrids, along with their antimicrobial and antiviral activities, as well as the structure–activity relationship reported in the literature. It was found that the presence of certain groups grafted onto the benzimidazole and/or triazole nuclei (-F, -Cl, -Br, -CF_3_, -NO_2_, -CN, -CHO, -OH, OCH_3_, COOCH_3_), as well as the presence of some heterocycles (pyridine, pyrimidine, thiazole, indole, isoxazole, thiadiazole, coumarin) increases the antimicrobial activity of benzimidazole-triazole hybrids. Also, the presence of the oxygen or sulfur atom in the bridge connecting the benzimidazole and triazole rings generally increases the antimicrobial activity of the hybrids. The literature mentions only benzimidazole-1,2,3-triazole hybrids with antiviral properties. Both for antimicrobial and antiviral hybrids, the presence of an additional triazole ring increases their biological activity, which is in agreement with the three-dimensional binding mode of compounds. This review summarizes the advances of benzimidazole triazole derivatives as potential antimicrobial and antiviral agents covering articles published from 2000 to 2023.

## 1. Introduction

Heterocyclic compounds have a central place in medicinal chemistry, being used as therapeutic agents to treat most diseases [1,2,3]. Among these heterocycles, benzimidazole stands out, as a purine-analog pharmacophore, with a very diverse therapeutic activity. The very broad spectrum of biological activities it treats include antimicrobial [4,5,6,7,8], antiviral [9,10], antihistamine [11,12], anticonvulsant [3,13], antitumor [14,15,16], proton pump inhibitors [17], antiparasitic [16,18,19], anti-inflammatory [20,21,22], or antihypertensive [23,24] activities. Some benzimidazoles are efficient agents in Diabetes mellitus [25,26,27], while astemizole compounds possess anti-prion activity to treat Creutzfeldt-Jakob disease [5,28]. The literature also reports anti-Alzheimer [29,30], psychoactive, anxiolytic, analgesic [31,32], and anticoagulant properties [33,34] of benzimidazole derivatives.

Additionally, triazole compounds possess a diversity of biological activities as antimicrobial [35,36,37,38], antitubercular [39,40], potential inhibitors of SARS-CoV-2 [41,42,43], antiviral [43,44], anti-inflammatory [45,46],antitumor [47,48,49,50], antihypertensive [50], antioxidant [47,51,52], and antiepileptic [53,54]. Pharmacological applications of triazoles refer to their activity as α-glucosidase inhibitors [55,56], analgesics [50,57], anticonvulsants [53,58], and antimalarial agents [57,59]. Triazole derivatives are efficient in the treatment of Alzheimer’s disease [60,61] and are very effective neuroprotective agents [62,63].

The successive events that occurred from the spring of 2020 until now, regarding the emergence and development of the COVID-19 pandemic, have led the scientific world to investigate more closely the possibility of treating this infectious disease with various antiviral [64,65,66], antimicrobial [67], immunomodulatory [68] or anti-inflammatory [69] drugs, therefore, the discovery of new molecules with simple or hybrid structures, which meet the requirements of the treatment of this condition is absolutely necessary and constitutes the engine for the development of new effective therapeutic agents.

Why did I choose the study of benzimidazole-triazole compounds? Classical drugs containing benzimidazole and triazole rings recommend these heterocycles as essential in building new target compounds with antimicrobial, antiviral, antiparasitic, etc. properties (Figure 1). In addition, the literature mentions a series of benzimidazole-triazole hybrids with remarkable antimicrobial properties, and antiviral activities, including new anti-SARS-CoV-2 agents [70,71,72,73,74], with particular importance in the context of the recent pandemic, which led to the study of synthesis methods, antimicrobial properties, structure–property relationships, and their biological activities.

Therefore, this review aims to provide an update on the synthesis methods of the benzimidazole-triazole hybrids, along with their antimicrobial and antiviral activities, as well as the structure–activity relationship and DFT studies reported in the literature. The advantages of the study of benzimidazole-triazole hybrid compounds refer to a wider range of antimicrobial activities, compared to simple precursor heterocycles, to their better minimum inhibitory concentrations compared to simple component heterocycles, as well as to the need to hire specialized personnel to carry out this research.

The main disadvantages are material because the synthesis of some hybrid compounds requires high costs compared to simple heterocycles, as well as greater time consumption. Consequently, if the synthesized hybrids have increased biological properties compared to simple precursor heterocycles, the balance clearly tilts towards the advantage of the synthesis of hybrid compounds. However, access to hybrid compounds will not be without both sides, advantages and disadvantages, which requires careful prospecting of all the components involved in the production of hybrids.

As expected, the literature mentions benzimidazole-triazole hybrids with other biological properties than those studied in this review, such as antitumor [15,48,75,76,77,78,79,80,81,82,83,84,85,86,87,88,89,90,91,92], antioxidant [93,94,95], anti-Alzheimer [96,97,98,99], antidiabetic [100,101,102,103,104], and anti-inflammatory [105] properties, which is additional proof of the therapeutic potential of these hybrids and the need to study these hybrids on the topic proposed in the title. As expected, the study refers to both 1,2,3-triazole-benzimidazole hybrids and 1,2,4-triazole-benzimidazole hybrids, even if it seems that the literature is richer in the second category in terms of antimicrobial activity.

The recent literature marks several strategies for the synthesis of 1,2,3-triazoles, like click reaction [106], Bouiton-Katritzky rearrangement [107], oxidative cyclization of hydrazones [108], post-cycloaddition functionalization [109], alkylation or arylation of triazoles [110]. Also, for benzimidazoles, the literature mentions several methods of synthesis, such as the reaction of *o*-phenylenediamine with aldehydes or ketones (Phillips-Ladenburg reaction) [3,111,112,113], with acids or their derivatives (Weidenhagen reaction) [81], or green methods of classic syntheses [111,114,115,116,117].

Why this review is necessary and what exactly it proposes I will clarify in what follows. This article summarizes for the first time in the literature: various synthesis methods of benzimidazole-1,2,3-triazole hybrids as well as benzimidazole-1,2,4-triazoles, their antimicrobial and antiviral activities, as well as SAR studies and DFT performed on the mentioned hybrids. Where necessary, for compounds with superior biological activities, several examples from the literature were given, and the various studies performed on them (in vitro, in vivo, in silico, etc.) were mentioned. All of these aim at directing the syntheses of hybrid compounds with specific structures and superior antimicrobial and antiviral properties, taking into account the mentions reported in the literature up to now.

The database search methodology used in this review was the use of keywords, which can be found in the title, such as benzimidazole, 1,2,3-triazole, click reaction, 1,2,3-triazole, benzimidazole-triazole hybrids, antimicrobial, antiviral, or therapeutic properties, in different websites, such as PubMed, MDPI, Science Direct, Springer, The Royal Society Chemistry, ACS Publications, and Taylor & Francis. The selection of scientific articles for the last ten years was made according to the novelty brought in the benzimidazole-triazole hybrids and their antimicrobial and antiviral properties, as well as the therapeutic properties of the reported compounds.

Generally, articles from the last ten years have been selected. For the hybrids found, first, the syntheses and then their biological properties were presented, with special emphasis on those with improved properties (active on a larger range of microbial strains, with better minimum inhibitory concentrations, or where SAR studies were performed, DFT, etc.). In the following, we will present syntheses of benzimidazole-triazole hybrids with antimicrobial and antiviral properties. In order to highlight the structures of the heterocycles in the discussed compounds, we colored the benzimidazole nucleus with red, 1,2,3-triazole with blue, and 1,2,4-triazole with green.

## 2. Synthesis and Antimicrobial Activities of Benzimidazole-1,2,3-Triazoles

### 2.1. 2-Benzimidazole-R(Ar)-1,4-Disubstituted-1,2,3-Triazole Hybrids

Two series of new hybrids, 2-[4-((1*H*-benzimidazol-2-ylthio)methyl)-1*H*-1,2,3-triazol -1-yl]N′-(arylmethylidene)acetohydrazides (**2a**–**2l**) and 2-[4-((1*H*-benzimidazol-2-ylthio) methyl)-1*H*-1,2,3-triazol-1-yl]N-(α-arylethylidene)acetohydrazides (**3a**–**3f**) were prepared by Youssif et al. in two steps starting from 2-[4-((1*H*-benzimidazol-2-ylthio) methyl)-1*H*-1,2,3-triazol-1-yl] acetohydrazide **1** (Figure 1). Compounds **2a**–**2l** exhibited pronounced antibacterial activity, which ranged from 35 to 75% of that of the standard drug against *Staphylococcus aureus* and 50–80% of that of Ciprofloxacin against *E. Coli* (MIC values of 3.125–12.5 μmol mL^−1^). Compound **2k** showed the highest activity against *S. aureus* (75% activity, MIC = 12.5 μmol mL^−1^), while compound **2d** was the most active derivative against *E. Coli* (80% activity, MIC = 3.125 μmol mL^−1^). All the synthesized compounds were tested as potential antifungal agents against *Candida albicans* using Fluconazole as a reference drug. Compound **1** showed the activity of 48% of that of Fluconazole (MIC = 12.5 μmol mL^−1^). Compounds **2e** and **2k** displayed higher antifungal activity among the other derivatives as they showed 75% activity of that of Fluconazole (MIC = 3.125 μmol mL^−1^). Compounds **3a**–**3f** exhibited moderate to good activity against *E. Coli*, and their activity was 50–70% of that of Ciprofloxacin (MIC values of 6.25–12.5 μmol mL^−1^), and compounds **3a** and **3f** were the most active compounds against *E. coli* as they showed 70% of that of Fluconazole (MIC = 6.25 μmol mL^−1^) while compound **3b** showed the highest activity against *Staphylococcus aureus* (65% of that of Ciprofloxacin, MIC = 18 μmol mL^−1^) [118]. Al-blewi et al. used an azide–alkyne Huisgen cycloaddition reaction carried out by simultaneously mixing thiopropargylated benzimidazole **4** with the appropriate sulfa drug azides **5a**–**5f**, copper sulfate, and sodium ascorbate in DMSO/H_2_O to regioselectively furnish target mono-1,4-disubstituted-1,2,3-triazole tethered benzimidazole-sulfonamide conjugates **6a**–**6f** with 85–90% yields after 6–8 h of heating at 80 °C (Figure 2). All compounds were evaluated for their antimicrobial activity (Table 1) against four pathogenic bacterial strains (Gram-positive: *Bacillus cereus* ATTC 10876, *Staphylococcus aureus* ATTC 25,923, and Gram-negative: *Escherichia coli* ATTC 25922, *Pseudomonas aeruginosa* ATTC 27,853 and two fungal strains, *Candida albicans* ATTC 50193, *Aspergillus brasiliensis* ATTC 16404). As can be seen in Table 1, compound **6a** showed the best antibacterial activity against *Bacillus cereus* and *Staphylococcus aureus* (64 μg mL^−1^), and compounds **6c**, **6d**, and **6e** showed the best antibacterial activity against *Escherichia coli* (64 μg mL^−1^) [119]. Evaluation of in silico physicochemical properties or ADMET (adsorption, distribution, metabolism, excretion, and toxicity) as a robust tool to confirm the potential of a drug candidate was applied for these compounds [120]. As per Lipinski’s rule of five, an orally administered drug should have a log *p* ≤ 5, a molecular weight (MW) < 500 Daltons, and an HBD ≤ 5 [121] to be in the acceptable range. Results have shown that all hybrids have in good agreement in terms of HBD. Rashdan et al. synthesized hybrids **10** starting from 2-azido-1*H*-benzo[d]imidazole derivatives **7a**–**7b**, which reacted with acetylacetone in the presence of sodium ethoxide to obtain hybrids molecules **8a**–**8b**. The latter acted as a key molecule for the synthesis of new carbazone derivatives **9a**–**9b** that were submitted to react with 2-oxo-N-phenyl-2 (phenylamino)acetohydrazonoyl chloride to obtain the target hybrid derivatives **10a**–**10b** (Figure 3). All compounds were screened for their in vitro antimicrobial activity against pathogenic microorganisms *Staphylococcus aureus*, *E. coli*, *Pseudomonas aeruginosa*, *Aspergillus niger*, and *Candida albicans*. The results showed that compounds **10a** and **10b** had strong activity against all the tested pathogenic microbes. Compounds **8a** and **9a** only showed effects against the Gram-negative and Gram-positive bacteria and had no effect on the tested fungi. In addition, in silico, and in vitro findings showed that compounds **10a** and **10b** were the most active against bacterial strains and could serve as potential antimicrobial agents (Table 2). The hybrids **8**–**10** were subjected to molecular docking studies with DNA gyrase B and exhibited binding energy that extended from -9.8 to -6.4 kcal/mol, which confirmed their excellent potency. The compounds **10a** and **10b** were found to be with the minimum binding energy (−9.8 and −9.7 kcal/mol) as compared to the standard drug Ciprofloxacin (−7.4 kcal/mol) against the target enzyme DNA gyrase B, as summarized in Figure 2 [122].

ADMET analysis of compounds **8**–**10** exhibited that they have good absorption properties (%HIA) ranging from 99.57 to 100% [123]. For distribution, the compounds do not permeate the blood–brain barrier (BBB). Moreover, the molecules were negative in the AMES toxicity and carcinogenicity test, which suggests that they are non-mutagenic. Compounds **11a**–**11g** with terminal acetylene and 2-(azidomethoxy)ethyl acetate were condensed using CuI as catalyst and triethylamine (TEA) under microwave irradiation to achieve hybrids 1,2,3-triazole connected via benzene to the benzimidazole nucleus **12a**–**12g** with excellent yields (70–90%) (Figure 2). The cleavage of the acetyl group using potassium carbonate (K_2_CO_3_) in methanol liberated the hydroxy group of the corresponding hybrid triazoles **13a**–**13g** (Figure 4). in almost quantitative yields. Compounds **13a**–**13g** were screened for in vitro antifungal activities against two phytopathogenic fungi, *Verticillium dahliae* Kleb and *Fusarium oxysporum* f. sp. *albedinis*. The result of the mycelia linear growth rate indicates that some of the compounds show a weak inhibition against the two fungi, the only compound that shows a significantly increased rate is compound **13e**, with a rate of 29.76% against *Verticillium dahliae* in the sporulation test [124].

Bistrović et al. synthesized in two steps hybrids **19a**–**19e**, **20a**–**20e**, and **21a**–**21e** starting from 4-(prop-2-ynyloxy)benzaldehyde **14** (Figure 5). All compounds were evaluated for their in vitro antibacterial activity against Gram-positive bacteria: *S. aureus* ATCC 25923, methicillin-sensitive *S. aureus*, *E. faecalis*, vancomycin-resistant *E. faecium*, and Gram-negative bacteria: *E. coli* ATCC 25925, *P. aeruginosa* ATCC 27853, *A. baumannii* ATCC 19,606 and ESBL-producing *K. pneumoniae* ATCC 27736. Generally, compounds showed better activities against Gram-positive than Gram-negative bacteria. Compounds **20a**–**20e**, with better binding affinity relative to other amidines, were the most active against *S. aureus* (MIC = 8–32 µgmL^−1^). Compound **19a** was the most promising candidate because of its higher potency (MIC = 4 µgmL^−1^) against ESBL-producing *E. coli*. Results of anti-trypanosomal evaluations showed that *p*-methoxyphenyl substituent in **19b**–**21b** enhanced activity, with **20b** (IC_50_ = 1.1 mM and IC_90_ = 3.5 mM) being more potent than Nifurtimox. In contrast to the observed correlation between antimicrobial activity and DNA binding, the antiprotozoal effects of **20b** did not correlate with its DNA affinity [125].

Rao et al. synthesized hybrids **22a**–**22b** (Figure 3) using the click chemistry approach. Compounds had weak activity against *Mycobacterium bovis* strain, with BCG values % inhibition = 27.3 and 26.2, respectively, at 30 µM concentration, using a turbidometric assay. Compounds **22a** and **22b** also showed moderate antiproliferative activity against human breast cancer cell line MCF-7, with IC_50_ values of 31.9 and 25.1 μM, respectively [126]. Ashok et al. synthesized in three steps hybrids **26a**–**26j**, starting from 1*H*-indole-3-carbaldehyde **7** (Figure 6). The compounds were evaluated for their antimicrobial activity against Gram-positive *Staphylococcus aureus* ATCC 6538, *Bacillus subtilis* ATCC 6633, and Gram-negative *Proteus vulgaris* ATCC 29213, *Escherichia coli* ATCC 11,229 bacteria using Gentamicin as standard. Antifungal activity was tested against *Candida albicans* ATCC 10,231 and *Aspergillus niger* ATCC 9029 strains with the standard drug Fluconazole. Compounds **26b**, **26c**, and **26h** with MIC of 3.125–6.25 μg mL^−1^ were found to be the most promising potential antimicrobial molecules [127]. The authors calculated various physicochemical parameters such as clogP, drug score, and drug-likeness of **26a**–**26j** using the Osiris Property Explorer software [128]. For all the compounds, the calculated clogP values were found to be below five according to Lipinski’s rule-of-5 and also exhibited positive values for drug score. Mallikanti et al. synthesized novel benzimidazole-conjugated 1,2,3-triazole analogs **29a**–**29l** in two steps: 1. formation of benzimidazole intermediate by reaction between 3’,5’-difluorobiphenyl-3,4-diamine **27** and 2-hydroxy-4-(prop-2-ynyloxy) benzaldehyde **28**, and 2. microwave-assisted copper-catalyzed click reaction (Figure 7). Compounds **29a**–**29l** showed minimal inhibition zones against all Gram-positive (*S. aureus*, *B. subtilis*) and Gram-negative (*E. coli*, *P. aeruginosa*) strains using Ampicillin as A standard drug. Among all tested compounds, the **29i** and **29k** showed higher activity against *P. aeruginosa*, *S. aureus*, and *B. subtilis* than the standard reference. Compounds **29a**, **29b**, **29c**, **29d**, **29e**, **29f**, **29g**, **29h**, **29j**, and **29l** showed moderate antibacterial activity against tested strains (Inhibition zone: 10–25 mm compared with 18–20 mm for Ampicillin). Compounds **29i**, **29j**, and **29k** also established strong activity against both fungal strains, *C. albicans* MTCC 183 and *A. niger* MTCC 9652, compared to the standard drug Griseofulvin [70]. To understand the binding mode of novel compounds, docking simulations were performed against the crystal structures of glucosamine-6-phosphate synthase (GlmS) (PDB ID: 2VF5) of *E. coli* and secreted aspartic proteinase (Sap) 1 (PDB ID: 2QZW) of *C. albicans* retrieved from the protein data bank. The best active compound, **29l**, scored the highest binding affinity value of about −10.0 kal/mol, which demonstrated two key interactions with the active site amino acid Asp549 of Glms with a bond distance of 2.66 and 2.81 Å. Further, the hydrophobic interactions were taken with Tyr312, Ser316, Asp474, Asn523, Ala572, and Ala551 of Glms, among which one π-π T-shaped interaction with Tyr312, and halogen bond [129] interactions with Tyr312, Asn523, and Asn551 (Figure 4). The binding energies and interactions of all compounds are better than that of Ampicillin, which proves that these molecules could best fit into the cavity of Glms [70]. Chandrika et al. reported hybrids **30**–**32** with broad spectrum antifungal activity (0.975–3.9 µgmL^−1^ against *C. albicans*; 0.12–0.48 µgmL^−1^ against *C. parapsilosis*) (Figure 5). These compounds also displayed good activity against *C. albicans* biofilms (3.9–15.6 µgmL^−1^ against *C. albicans*) [130].

### 2.2. 1-Benzimidazole-R(Ar)-1,4-Disubstituted-1,2,3-Triazole Hybrids

Deswal et al. synthesized a new series of benzimidazole-1,2,3-triazole-indoline derivatives **35** by employing a click reaction between substituted N-propargylated benzimidazole derivatives **33** and in situ formed substituted 2-azido-1-(indolin-1-yl) ethanone derivatives **34**, in moderate to good yields (Figure 8). The obtained results indicate a stronger inhibitory effect of compound **35d** against *E. coli*, while compound **35g** showed good inhibition against *all* the tested strains except *B. subtilis* (Table 3). The good antimicrobial activity of the compounds was correlated with the presence of the pyridine ring in position ‘‘2’’ of the benzimidazole and the NO_2_ group on the indole ring. Furthermore, in vitro α-glucosidase inhibition of all synthesized derivatives identified **35e** (IC_50_ = 0.015 ± 0.0003 μmol mL^−1^) and **35g** (IC_50_ = 0.018 ± 0.0008 μmol mL^−1^) as potent inhibitors of α-glucosidase, even better than standard drug Acarbose [6].

Saber et al. synthesized new 1,4-disubstituted-1,2,3-triazole containing benzimidazolone derivatives **37a**–**37d** exclusively using click chemistry (Figure 9). All derivatives exhibited antibacterial activity against tested strains, *Staphylococcus aureus*, *Escherichia coli,* and *Pseudomonas aeruginosa*, but compounds **37b** and **37d** are more effective against Gram-positive bacterium *S. aureus* (MIC = 3.125 µgmL^−1^), and **37b** has better activity against Gram-negative bacterium *E. coli* (MIC = 3.125 µgmL^−1^) with Chloramphenicol as standard drug. The expected inhibition efficiency, **37c** > **37a** > **37b**, was attributed to the favorable effect of the side carbon chain of the triazole moiety, according to DFT calculations, in this process [131,132]. Mohsen et al. synthesized hybrids **41a**–**41e** in three steps, starting from benzimidazole **38**, namely two alkylation reactions and a click reaction (Figure 10). New derivatives exhibited good zone inhibition of 6.8, 5.4, 5.2, 4.5, and 5.3 mm for the *S. aureus* strain and 5.4, 3.8, 4.2, 3.3, 4.9 mm for the *E. coli* strain, indicating that the 1,2,3-triazole core contributed significantly to bacterial growth suppression (Ciprofloxacin showed 10.2 mm for *S. aureus* and 10.4 mm for *E. coli*). Compared with Gram-negative bacteria, all compounds showed a strong effect against Gram-positive bacteria [94].

### 2.3. 1,2-Bis-Substitutedbenzimidazoles-R(Ar)-1,4-Disubstituted-1,2,3-Triazole

Rezki reported the intramolecular cyclization of thiosemicarbazides **42a**–**42d** in refluxing aqueous sodium hydroxide (2N) for 6 h with the formation of hybrids **43a**–**43d** with yields of 82–86% (Figure 11). Among all the 1,2,4-triazole derivatives, N4-phenyl and N4-(4-fluorophenyl) derivatives **43a** and **43b** were found to be the most potent with MIC values of 4–8 µg mL^−1^. Also, triazoles **43c** and **43d** exerted the best inhibition against both tested fungal strains, *A. brasiliensis* and *Candida albicans*, with MIC values ranging from 0.5 to 4 µg mL^−1^, more potent than the reference drug Fluconazole. Condensation of compound **44** with several benzaldehydes in refluxing ethanol for 4–6 h with a catalytic amount of HCl produced a new class of hybrid Schiff bases **45a**–**45g** with yields of 84–86% (Figure 12). The antimicrobial bioassay results for the synthesized Schiff bases **45a**–**45g** revealed that all of the tested compounds were more effective towards all of the organisms, with MIC values of 1–16 µg mL^−1^. Among them, Schiff bases **45c**, **45d**, and **45e** with a fluorine atom at position “2” exhibited the highest antibacterial inhibition potency at MIC 1–8 µg mL^−1^. The Schiff base **45e** containing a CF_3_ group exerted the highest antifungal inhibition activity with MIC of 1 µg mL^−1^ [133]. Al-blewi et al. synthesized triazoles **47a**–**47f** in two steps: i. regioselective alkylation of **4** with two equivalents of propargyl bromide in the presence of two equivalents of potassium carbonate as a base catalyst to afford benzimidazole **46** with 91% yield after stirring at room temperature overnight; ii. Copper-mediated Huisgen 1,3-dipolar cycloaddition reaction on compound **46** in good yields (82–88%) (Figure 13). Generally, bis-1,2,3-triazoles **47a**–**47f** exhibited more potent antimicrobial activities than their mono-1,2,3-triazole derivatives **6a**–**6f**. This was attributed to the synergistic effect of the sulfonamoyl and tethered heterocyclic components in addition to the improved lipophilicity of the bis-substituted derivatives. Among the synthesized compounds, compound **47a** was the most potent antimicrobial agent, with MIC values ranging between 32 and 64 μg mL^−1^ against all tested strains *B. cereus*, *S. aureus*, *E. coli P. aeruginosa*, *C. albicans*, and *A. brasiliensis*. Pharmacophore elucidation of the compound **47a**–**47f** was performed based on in silico ADMET evaluation of the tested compounds. Screening results of drug-likeness rules showed that all compounds follow the accepted rules, meet the criteria of drug-likeness, and follow Lipinski’s rule of five. In addition, the toxicity results showed that all compounds are non-mutagenic and noncarcinogenic [119].

Aparna et al. used a similar strategy for obtaining nine new bis-1,2,3-triazol-1*H*- 4-yl-substituted arylbenzimidazole-2-thiol derivatives **48a**–**48l** (Figure 6). Antibacterial activity of triazole derivatives **48** demonstrates moderate to good activity against Gram-negative (*E. coli*, *S. typhy*, *P. aeruginosa*) and Gram-positive (*S. aureus)* bacterial strains. The products **48i**, **48k**, and **48l** are characterized by a broad spectrum of antibacterial activity at a concentration of 10 μg mL^−1^. The synthesized 1,2,3 triazole derivatives were studied for their molecular docking on the high-resolution X-ray crystal structure of FabI of *Staphylococcus aureus* (pdb id:4FS3) obtained from the protein data bank [134]. The highest dock score of –7.69 kcal/mol and the lowest dock score of –0.942 kcal/mol were obtained for molecules **48l** and **48h**, respectively [135].

### 2.4. Benzimidazole-R(Ar)-1,2,3-Triazole Hybrids as Antitubecular Agents

Ashok reported compound **26h** has the best antitubercular drug candidate by inhibiting the growth of the MTB (*Mycobacterium tuberculosis*) strain with MIC = 3.125 μ mL^−1^ (7.1 μM) (control Rifampicin MIC = 0.04 μg mL^−1^ and isoniazid MIC = 0.38 μg mL^−1^). The best antitubercular activity of **26h** may be attributed to the presence of the nitro group on the phenyl ring at the *ortho* position. Compound **26b** (MIC = 6.25 μg mL^−1^ (14.7 μM)) with chlorine substituent, compound **26i** (MIC = 6.25 μg mL^−1^(14.2 μM)) with trifluoromethyl substituent and compound **26j** (MIC = 12.5 μg mL^−1^ (28.4 μM)) with benzyl substituent exhibited moderate antitubercular activity. Therefore, the incorporation of the electron-withdrawing nitro group, electronegative chlorine, and trifluoromethyl groups on the phenyl ring was highly favored for antitubercular activity. The authors calculated various physicochemical parameters and found from the theoretical data that compounds **26a**–**26j** also exhibited positive values for drug score [127]. Gill et al. reported syntheses of hybrids **51a**–**51d** by reaction between 2-(3-fluorophenyl)-1*H*-benzo[d]imidazole **50** and phenyl-substituted 4-(bromomethyl)-1-phenyl-1*H*-1,2,3-triazole **49** in DMF at room temperature (Figure 14). Trifluorosubstituted-compound **51a** possessed enhanced anti-mycobacterial activity, >96% of inhibition at 6.25 µg concentration. Also, compounds **51b** and **51c**, which had antimicrobial activities superior to the other compounds, were reported as the best choice for the preparation of new derivatives in order to improve effectiveness on intracellular mycobacteria (macrophage) or in infected animals [136]. Anand et al. reported a one-pot reaction between 2-propargylthiobenzimidazole **4**, 4-bromomethylcoumarins/1-aza-coumarins **52**/**53** and sodium azide under click chemistry conditions to give exclusively 1,4-disubstituted triazoles **54a**–**54n**. (Figure 15). Antitubercular assays against *M. tuberculosis* (H37Rv) coupled with in silico molecular docking studies indicated that dimethyl substituents **54c** and **54d** showed promising activity (MIC = 3.8 µMol L^−1^) with higher C-score values. Surflex-Dock was used to investigate detailed intermolecular interactions between the ligand and the target protein. Three-dimensional structure information on the target protein was taken from the PDB entry 4FDO. Processing of the protein included the removal of the co-crystallized ligand and water molecules, as well as the addition of essential hydrogen atoms. All 14 inhibitors **54a**–**54n** were docked into the active site of ENR, as shown in Figure 7a, and Figure 7b indicates the superimposition of compounds **54a** and **54d** with ligand [137]. Khanapurmath et al. synthesized triazoles **55** by click reaction (Figure 8a). Benzimidazolone bis-triazoles **55a**–**55n** showed better activity with MIC in the range 2.33–18.34 μM, and the most active compounds were **55h** and **55m**. All compounds exhibited moderate to low levels of cytotoxicity with IC_50_ values of the human embryonic kidney cells in the range of 943–12294 μM, and none of the 14 compounds exhibited any significant cytotoxic effects, suggesting huge potential for their in vivo use as antitubercular agents. Docking studies revealed an additional interaction of benzimidazolone oxygen in these compounds (Figure 8b) [138]. Also, Sharma et al. summarize 1,2,3-triazoles as antitubercular compounds and various hybrids with benzimidazole, coumarin, isoniazid, quinolines, etc. [39].

## 3. Synthesis and Antimicrobial Activities of Benzimidazole-1,2,4-Triazoles

### 3.1. 2-Benzimidazole-R(Ar)-1-(1,2,4-Triazole)

Pandey et al. synthesized hybrids **59a**–**59e** in three steps: reaction of 7-hydroxy-4-methyl coumarin with thiosemicarbazide to form triazole intermediate 57, which underwent Mannich reaction with formaldehyde, and an amino acid to form intermediates **58a**–**58e**. Intermediates **58a**–**58e** reacted with *o*-phenylenediamine in pyridine to give benzimidazole-1,2,4-triazole hybrids in poor yields (Figure 16). Compound **59a** displayed promising antifungal activity against *Candida albicans* and *Cryptococcus himalayensis* since the MIC value in each case was found to be 3.5 μg mL^–1^. Compound **59b** showed low to moderate antifungal activity against all five fungi, *Candida albicans*, *Cryptococcus himalayensis*, *Sporotrichum schenkii*, *Trichophyton rubrum*, and *Aspergillus fumigatus* [139].

Jadhav et al. synthesized a series of hybrids 1,2,4-triazolyl-fluorobenzimidazoles in two steps: i. synthesis of 2-(4-(1*H*-1,2,4-triazol-1-yl)phenyl)-4,6-difluoro-1*H*-benzo [d]imidazole **62** by reaction between 3,5-difluorobenzene-1,2-diamine **60** and 4-(1*H*-1,2,4- triazol-1-yl)benzaldehyde **61** in toluene at 110 °C, and ii. alkylation of compound **62** in DMF at room temperature, with the formation of the final hybrids **63a**–**63o** (Figure 17). All compounds were screened for antimicrobial activity against different Gram-positive organisms, *S. aureus*, *P. aeruginosa*, and Gram-negative organisms, *E. coli* and *S. typhosa* using Gentamycin as a reference standard. The data generated from preliminary screening showed that compounds displayed moderate to better antimicrobial activity. Compounds **63a**, **63e**, **63f**, **63h**, **63i**, and **63l** displayed maximum activity (Table 4) [140].

Barot et al. synthesized hybrid **64** and determined its antimicrobial activity against *Bacillus cereus* MTCC-430, *Enterococcus faecalis* MTCC-493, *S. aureus* MTCC-737, *Escherichia coli* MTCC-1687, *Pseudomonas aeruginosa* MTCC-2642, *Klebsiella pneumonia* MTCC-109, *Candida albicans* MTCC-3017, *Aspergillus niger* MTCC-1344 and *Fusarium oxyspora* MTCC-1755, of MIC = 13–18 µg ml^−1^, with Ofloxacine and Fluconazole as standard drugs [141]. Also, Jiang et al. reported antifungal activity for hybrid **65** against *Candida albicans*, *Candida tropicalis*, *Cryptococcus neoformans*, *Trichophyton rubrum*, and *Aspergillus fumigatus* of MIC_80_ = 1–64 µg mL^−1^ considering Fluconazole as a standard drug (Figure 9). From the antifungal activity data, preliminary SARs was obtained. In general, the amine linker was important for antifungal activities. Substituted piperazine derivatives were comparable or superior to the corresponding N-methyl derivatives. [142]. Luo et al. reported a series of naphthalimide benzimidazole-1,2,4-triazole hybrids **68a**–**68h** and the corresponding triazolium salts **69a**–**69d** prepared by convenient and efficient procedures starting from naphthalimide triazole **66** (Figure 18). 2-Chlorobenzyl triazolium **68g** and compound **69b** with octyl group exhibited the best antibacterial activities among all the tested compounds, especially against *S. aureus* with an inhibitory concentration of 2 μg mL^−1^ which was equipotent potency to Norfloxacin (MIC = 2 μg mL^−1^) and more active than Chloromycin (MIC = 7 μg mL^−1^). Triazoliums **68g** and **68f** bearing 3-fluorobenzyl moiety displayed the best antifungal activities (MIC = 2–19 μg mL^−1^) against all the tested fungal strains, *C. albicans* ATCC 76615, *A. fumigatus* ATCC 96918, *C. utilis*, *S. cerevisia* and *A. flavus*, without being toxic to PC12 cell line within concentration of 128 μg mL^−1^. Further investigations showed that compound **68g** could intercalate into calf thymus DNA to form the **68g**-DNA complex, which could block DNA replication, exerting powerful antimicrobial activities. [143]. Benzimidazole-1,2,4-triazole Mannich base **70** was active against *Bacillus subtilis* and *Bacillus pumilus* (inhibition zone diameters being 19 and 17 mm, respectively, compared to Ciprofloxacin with 28 and 30 mm, respectively) [144].

Kankate et al. reported the synthesis of hybrids **73a**–**73l** (Figure 19). The antifungal activity of compound **73** was tested against *Candida albicans* spores in vitro (turbidimetric method) and in vivo (kidney burden test). Compound **73i** had a good antifungal activity as compared with the other twelve compounds at 0.0075 µM mL^−1^, which is equivalent to Fluconazole activity both in vitro and in vivo. The antifungal activity decreased with the increasing alkyl length of N1 of benzimidazole (methyl to ethyl). This was proven for compounds **73i** and **73l**, which showed MICs of 0.0075 and 0.015 µM mL^−1^, respectively. The ligand fit method was performed to study and predict the binding mode of the hybrids **73** with the target enzyme (homology modeled) cytochrome P450 lanosterol 14-α-demethylase of *C. albicans*. All compounds showed binding in the active site of the enzyme. The 1,2,4-triazole ring of compounds **73a**–**73l** is positioned almost perpendicular to the porphyrin plane, with a ring nitrogen (N-4) atom coordinated to the heme iron (Figure 10) [145,146]. Ahuja et al. reported antifungal activity of compounds **74a**–**74c** against *F. verticillioides*, *D. oryzae*, *C. lunata,* and *F. fujikuroi* (Figure 11). All compounds had increased potency than the standard commercial benzimidazole fungicide, carbendazim (Table 5).

Compound **74c** exhibited ED_50_ values lower than triazole fungicide, propiconazole. The results reinforced the synergistic effects of the benzimidazole and 1,2,4-triazole combination supported by a computational approach. Hydrogen bonding interactions were more pronounced in compounds **74a**–**74c** in the binding pockets of both the target enzymes, in comparison to standards. In compound **74c**, two H-bonds were formed with Gln11 present in the binding cleft of the active pocket of β-tubulin (Figure 12). In all three compounds, the right position of the N-atoms of both 1,2,4-triazole and benzimidazole, that O-atoms of methoxy and carbonyl groups, contributed well to strong binding into the active site of enzymes via H-bonding [147]. Evren et al. reported the synthesis of the compounds **79a**–**79c** in two steps: i. reaction of 1,2,4-triazole **75** with 4-fluorobenzaldehyde **76** in DMF with the formation of 4-(1*H*-1,2,4-triazol-1-yl)benzaldehyde **77**; ii. reaction of aldehyde **77** with 1,2-phenylene diamines **78** (Figure 20). Although the antibacterial activities of compounds **79a**–**79c** against *Escherichia coli* ATCC 35218, *E. coli* ATCC 25922, *Klebsiella pneumoniae* NCTC 9633, *Pseudomonas aeruginosa* ATCC 27853, *Salmonella typhimurium* ATCC 13311, and *Staphylococcus aureus* ATCC 25923, were weak, the antifungal activities against *C. albicans* were found promising, with MIC values of 3.9, 7.8, and 3.9 μg mL^−1^ respectively, using as reference drug Ketokonazole (MIC = 7.8 μg mL ^−1^). Protein-ligand interactions and binding poses of the **79a**, **79b**, and **79c** compounds on the CYP51 active site were examined. As shown in Figure 13, an H bond of 2.11 Å between compound **79a** and Met508, π-π stacking interactions with Tyr118, Hie377, Phe233, generated hydrophobic interactions with Pro230, Leu376, Tyr64, Phe228, and Tyr505. Theoretical ADME calculations of the **79a**, **79b**, and **79c** were made, and the compounds were found to have good lipophilicity, moderate water solubility, and within the limiting rules of Lipinski, Ghose, Veber, Egan, and Muegge (Figure 14) [148]. Ghobadi et al. reported the synthesis of compounds **85a**–**85e**, in two different ways, from 3,4-diaminobenzophenone **80**, i. formation of 2-mercapto benzimidazole derivatives **82**, **83**, and ii. nucleophilic ring opening of various oxiranes **84a**–**84e** with benzimidazoles **82** and **83** using NaHCO_3_ in ethanol at room temperature (Figure 21). Compounds **85a**–**85e**, containing a 5-benzoylbenzimidazole scaffold, showed better antifungal activity against *Candida* spp. and *Cryptococcus neoformans* than related benzimidazole and benzothiazole derivatives. The better results were obtained with 4-chloro-derivative **85b** displaying MICs < 0.063–1 μg mL^−1^. Also, compound **86c**, synthesized analogously, is as potent as compound **85b**. The docking experiments were conducted to further rationalize the obtained antifungal activity data and investigate the type of interactions between compound **85b** and the active site of lanosterol 14α-demethylase (CYP51). As shown in Figure 13, the coordinated bond-forming distance between the N4 atom of the triazole nucleus of compound **85b** and the iron atom in the heme group of active site were 2.71 and 2.40 Å, respectively. A hydrogen-bonding interaction between Tyr132 and the sulfur group of (S)-**85b** was observed. In vitro and in silico ADMET evaluations of the most promising compounds **85b** indicated that the selected compounds have desirable ADMET properties in comparison to the standard drug Fluconazole. A docking simulation study demonstrated that the benzimidazol-2-yl-thio moiety is responsible for the potent antifungal activity of these compounds [72].

### 3.2. 1-Benzimidazole-R(Ar)-2-1,2,3-Triazole

Ansari et al. synthesized hybrids **88a**–**88c** in two steps from 2-(2-methyl-1*H*-benzo [d]imidazol-1-yl)acetohydrazide **87** (Figure 22). Generally, all benzimidazole-triazole hybrids showed low antimicrobial activity (Table 6) [149]. Tien et al. synthesized hybrids **89a**–**89d** in three steps from 2-(2-methyl-1*H*-benzo[d]imidazol-1-yl)acetohydrazide **87b** (Figure 23). All compounds exhibited antifungal activity against *A. niger* (MIC = 50 µg mL^−1^). Only compound **89b** exhibited activity against *F. oxysporum* (Table 7) [150]. Kantar et al. reported antimicrobial activity of hybrid **90** (Figure 15) against four Gram-positive, *Bacillus cereus* 702 Roma (62.5 µg mL^−1^), *B. megaterium* DSM-32 (125 µg mL^−1^), *B. subtilis* ATCC 6633 (62.5 µg mL^−1^), *Staphylococcus aureus* ATCC 25,923 (250 µg mL^−1^), and four Gram-negative bacteria, *Escherichia coli* ATCC 25,922 (250 µg mL^−1^), *Enterobacter cloaceae* ATCC13047 (125 µg mL^−1^), *Pseudomonas aeruginosa* ATCC 27,853 (250 µg mL^−1^), and *Yersinia pseudotuberculosis* ATCC 911 (125 µg mL^−1^) bacteria [151]. Nandwana et al. reported compound **91** synthesized in good yield (70%) with promising antibacterial activity, with minimum inhibitory concentration (MIC) values of 4−8 μg mL^−1^ for all bacterial tested strains (*Escherichia coli*, *Pseudomonas putida*, *Salmonella typhi*, *Bacillus subtilis*, *Staphylococcus aureus*), as compared to the positive control Ciprofloxacin, and also with pronounced antifungal activity against both tested strains, *Aspergillus niger* and *Candida albicans* (MIC = 8−16 μg mL^−1^) as compared with Amphotericin B [152]. Al-Majidi et al. synthesized 2-mercaptobenzimidazole derivatives **95**, **96**, and **97** by cyclization of intermediate precursors **93**, **94**, and **95** under reflux with 2N NaOH (Figure 24). The compounds generally showed moderate antimicrobial activity against all tested strains, as can be seen in Table 8 [153]. El-masry et al. synthesized compounds **98** and **99** and found that they did not exhibit antimicrobial activity (Figure 16) [154]. Menteşe et al. synthesized compounds **100a**–**100d**, for which they found no antimicrobial activity on the ten strains tested [155]. Karale et al. synthesized bis-benzimidazole-1,2,4-triazole hybrids **102a**–**102e** (Figure 25) in four steps from 7-methyl-2-propyl-3*H*-benzo[d]imidazole-5-carboxylic acid. All compounds **102** did not show antimicrobial activity against the strains tested, *C. albicans*, *A. fumigatus*, *S. aureus*, and *E. coli* [156,157].

### 3.3. 2-Benzimidazole-R(Ar)-2-1,2,4-Triazole

Eisa et al. synthesized compounds **105a** and **105b** (Figure 26) (Table 9) by the reaction between 2-(chloromethyl)-1*H*-benzo[d]imidazole **103** and 4-phenyl-5-(pyridin-3-yl)-4*H*-1,2,4- triazole-3-thiol **104a** or 4-phenyl -5-(thiophen-2-yl)-4*H*-1,2,4-triazole-3-thiol **104b**, at reflux in absolute ethanol, for 12 h. Also, they reported synthesis of the compounds **107a** and **107b** from 2-(2-(phenylthiomethyl)-1*H*-benzo[d]imidazol-1-yl)acetohydrazide in two steps (Figure 27). All compounds showed antimicrobial activity against *Escherichia coli* superior to that of standard Gentamicin. Compound **107a** exhibited only moderate activity against *Staphylococcus aureus* [158]. Nevade et al. synthesized compounds **109a**–**109h** in five steps from 1*H*-benzo[d]imidazole-2-thiol **108** (Figure 28). The antimicrobial screening results presented in Table 10 reveal that compounds **109a**, **109c**, and **109e** exhibited satisfactory effects against *S*.*aureus* and *E*.*coli*, while compounds **109b**, **109f**, and **109g** showed moderate activity against the same microbes. Also, the antifungal activity of these compounds was screened against *Candida albicans*. Compounds **109a** and **109d** showed the highest degree of inhibition against *C*.*albicans* when compared with the standard drug Ketoconazole [159]. Can et al. synthesized hybrids **111a**–**111h** in four steps from methyl 4-(5-methyl-1*H*-benzo[d]imidazol-2-yl)benzoate **110** (Figure 29). All compounds were screened for antifungal activity against *Candida albicans* ATCC 24433, *Candida glabrata* ATCC 90030, *Candida krusei* ATCC 6258, and *Candida parapsilosis* ATCC 22,019 (Table 11). Compounds **111i** and **111s** exhibited significant inhibitory activity against *Candida* strains with MIC_50_ values ranging from 0.78 to 1.56 μg mL^−1^ [160]. Gencer et al. synthesized compounds **112** in good yields (77–88%) using a similar strategy (Figure 17). Microbiological studies revealed that compounds **112a**, **112b**, **112c**, **112e**, **112f**, **112g**, and **112h** possess a good antifungal profile against all tested strains, *C. albicans*, *C. glabrata*, *C. krusei*, *C. parapsilopsis*, with MIC_50_ = 0.78–1.56 µg mL^−1^. Compound **112i** was the most active derivative and showed comparable antifungal activity to those of reference drugs Ketoconazole and Fluconazole [161]. The SAR (Structure–activity relationship) on the synthesized benzimidazole-triazole compounds is summarized in Figure 18. It is observed that the presence of chlorine or fluorine in the “5” position of benzimidazole, as well as the presence of fluorine in the “4” position of phenyl, increase the antibacterial activity, while the presence of fluorine in the “2” position of phenyl does not change the activity, and the presence of groups CH_3_ or C_2_H_5_ in position “4” in the triazole nucleus does not bring any change in the antibacterial activity of the compounds. Furthermore, toxicological and ADME studies indicated the relative potency of hybrids **112h** and **112i**, according to the literature [162,163,164,165,166]. Compound **112i** also inhibited ergosterol biosynthesis concentration dependently. Results of ergosterol level quantification assay and fluorescence microscopy studies revealed that the mechanism of action of hybrids is associated with the inhibition of ergosterol biosynthesis, which may subsequently result in altered membrane fluidity, plasma membrane biogenesis, and functions of fungi. Güzel et al. synthesized a new series of benzimidazole-1,2,4-triazole derivatives **113a**–**113l** using the same procedure described in Figure 29 as potential antifungal agents (Figure 19). All the compounds were screened for their in vitro antifungal activity against four fungal strains, namely, *C. albicans*, *C. glabrata*, *C. krusei*, and *C. parapsilopsis* and were found to exhibit excellent activity against *C. glabrata*. Especially, compounds **113b**, **113i**, and **113j** were found to be the most effective compounds in the series with an MIC value of 0.97 μg mL^−1^ [71]. According to the molecular docking study, compounds **113b**, **113i**, and **113j** fit into the LDM enzyme active pocket. In a previous study [167], the Tyr118 amino acid and HEM601 protein were described as essential residues, and in this study, the synthesized active compounds interacted significantly with Tyr118, His377, and HEM601 residues. The interactions with HEM were seen as π−π stacking and π−cation interactions. Therefore, the antifungal effects of compounds **113b**, **113i**, and **113j** were considered to be caused by the destruction of cell integrity due to the inhibition of the LDM enzyme. The authors identified compound **6i** with higher inhibitory activity due to H-bonding with Tyr132, unlike the other two compounds. Aryal et al. reported synthesis of 2-substituted benzimidazole containing 1,2,4-triazoles **114a** and **114b** (Figure 20). The compounds did not show antimicrobial activity against the tested strains *Staphylococcus aureus* ATCC 6538P and *Staphylococcus epidermidis* ATCC 1228 [168]. Kazeminejad et al. did a study on 1,2,4-triazoles as well as structure–activity relationships (SAR) [38].

### 3.4. 6-Substituted-Benzimidazole-R(Ar)-1-1,2,4-Triazole

Nandha et al. reported synthesis of 6-substituted-benzimidazoles with 1-(1,2,4-triazole) **115a**–**115d** in three steps from 5-chloro-4-fluoro-2-nitrobenzenamine (Figure 30). All compounds were screened against *M. tuberculosis* and four fungal strains, *C. albicans*, *C. glabrata*, *C. krusei*, and *C. tropicalis*. Compound **115c** was the most active against *M. tuberculosis* and all tested fungal strains (MIC = 25 μg mL^−1^) [169].

## 4. Synthesis and Antiviral Activities of Benzimidazole-Triazoles

Over 200 viruses are known to cause disease in humans, yet currently approved antiviral drugs are available to treat only about 10 of these viral infections [170,171]. The past decade has underscored the global threat posed by emerging viruses. An alternative solution is the development of broad-spectrum antiviral drugs. One advantage of this approach is reduced time and cost associated with the early stages of drug development per approved indication. It can also diminish the clinical risks in more advanced stages of development [172,173]. Youssif et al. reported the synthesis of benzimidazole-1,2,3-triazole hybrids 2-{4-[(1-benzoylbenzimidazol-2-ylthio)methyl]-1*H*-1,2,3-triazol-1-yl}-N-(4-nitro-phenyl)-acetamide **116** and 2-(4-{[1-(4-chlorobenzoyl)-benzimidazol-2-ylthio)methyl]-1*H*-1,2,3-triazol -1-yl}-N-(4-nitrophenyl)-acetamide **117** which showed significant activity against hepatitis C virus (HCV) (Figure 21). Thus, fifty percent effective concentrations (EC_50_) of HCV inhibition for compounds **116** and **117** were 7.8 and 7.6 μmol L^–1^, respectively, and the 50% cytotoxic concentrations (CC_50_) were 16.9 and 21.1 μmol L^–1^. The results gave an insight into the importance of the substituent at position 2 of benzimidazole for the inhibition of HCV [73].

The antiviral activity of compounds **59a**–**59e** was tested against two viruses, viz., *Japanese encephalitis virus* (JEV) (P20778), an RNA virus of higher pathogenicity, and *Herpes simplex virus* type-I (HSV-I) (753166), the most common virus present in the environment. The antiviral activity of the compounds data is given in Table 12. All but one of the five compounds were found active against JEV. Compound **59b** displayed 90% CPE (cytopathic effect) in vitro with an effective concentration of 8 µg mL^–1^, while in vivo activity was less significant (16% protection with an MST of 4 days). The authors suggested that these compounds are better anti-JEV agents than anti-HSV agents since two such compounds, namely **59b** and **59e**, also displayed a measurable degree of anti-JEV activity in vivo. Compound **59c** was found antivirally inactive against both viruses. The anti-HSV-I activity was found to be in the order of 33, 46, 53, and 64% for compounds **59a**, **59b**, **59d**, and **59e**, respectively. Since among compounds **59a** to **59e**, only compound **59e** contains a methyl group instead of H as R_1_; it follows that R_1_ does not seem to be responsible for the biological activity [139].

Tonelli et al. synthesized a series of 1-substituted 2-[(benzotriazol-1/2-yl)methyl] benzimidazoles **118**–**137** and tested for antiviral activity against a large panel of RNA and DNA viruses (Figure 22). Twelve compounds exhibited high activity against RSV (Respiratory Syncytial Virus), with EC_50_ values in most cases below 1 µM, comparing favorably with the reference drug 6-azauridine, which, moreover, exhibited high toxicity against both the MT-4 and Vero-76 cell lines (S.I. = 16.7). The observed activity against BVDV (Bovine Viral Diarrhea Virus), YFV (Yellow Fever Virus), and CVB2 (Coxsackie virus B2) is moderate, with EC_50_ values in the range of 6–55 µM for the best compounds (Table 13). Though not particularly impressive, the presently uncovered activity against BVDV, YFV, and CVB2 is of some interest because it may lead, through the identification of the target, to the development of broad spectrum antiviral agents. In this respect, the definition of the mode of action of the above compounds is mandatory. Furthermore, since the activity against these viruses was influenced by the presence and nature of the substituents in position “5” of the benzimidazole ring, it will be worthwhile to further explore the effect of diversified substitutions as a possibility to improve activity and/or decrease cytotoxicity [174]. SARS-CoV-2 and its variants, especially the Omicron variant, remain a great threat to human health [10]. More novel variants of SARSCoV-2 are also expected to originate in the future. Therefore, efforts should be made to develop wide-ranging measures to prevent future outbursts of zoonotic origin. Recent articles reported essential and up-to-date information about SARS-CoV-2 variants, antiviral drugs, and vaccines used to fight it [175,176].

Al-Humaidi et al. reported the synthesis of a series of benzimidazole-1,2,3-triazoles **138**–**140** (Figure 23). Molecular docking studies and in vitro enzyme activity revealed that most of the investigated compounds demonstrated promising binding scores against the SARS-CoV-2 and Omicron spike proteins in comparison to the reference drugs (Table 14).

Data proved the promising activity of the tested compound **140**, with its IC_50_ reaching 75.98 nM against the Omicron spike protein and 74.51 nM against the SARS-CoV-2 spike protein. The three-dimensional binding mode of compound **140** is shown in Figure 24. Benzimidazole-1,2,3-triazole hybrids can be potent anti-HSV (Herpes simplex virus) agents. These compounds were screened against flaviviruses and pestiviruses. Compound **141** showed excellent activity against respiratory syncytial virus (RSV) with an EC_50_ value of 0.02 mM (Figure 25) [74]. Seliem et al. designed and synthesized some quinolone–triazole conjugates against SARS-CoV-2. It was revealed that 4-((1-(2-chlorophenyl)-1*H*-1,2,3-triazol- 4-yl)methoxy)- 6-fluoro-2-(trifluoromethyl)quinoline and 6-fluoro-4-(2-(1-(4-methoxyphenyl) -1*H*-1,2,3- triazol-4-yl)ethoxy)-2-(trifluoromethyl)quinoline have high antiviral activity with a high selectivity index (SI) against SARS-CoV-2 in comparison to the reference drugs. They explained that the fluorine atoms in the tested compounds have a major role in the observed antiviral activity [43]. The importance of the 1,2,4-triazole ring in antiviral compounds is reviewed by El-Sebaey, who emphasizes the importance of the substituents in the triazole nucleus, as well as the important role of other heterocycles in the molecule [177].

## 5. Conclusions

This review summarizes the syntheses of benzimidazole–triazole compounds with antimicrobial and antiviral properties mentioned in the literature. The presence of certain groups grafted on the benzimidazole and triazole nuclei, such as -F, -Cl, -Br, -CF_3_, -NO_2_, -CN, -NHCO, -CHO, -OH, OCH_3_, -N(CH_3_)_2_, COOCH_3_, as well as other heterocycles in the molecule (pyridine, pyrimidine, thiazole, indole, isoxazole, thiadiazole, coumarin), increases the antimicrobial activity of the compounds [4,5,114,115,165,178,179,180]. From the presented literature data, we can highlight some aspects related to the correlation: structure—antimicrobial properties.

-The presence of substituents in the “4” or “5” positions of the benzimidazole nucleus can increase the antimicrobial activity of the benzimidazole-triazole hybrids (compounds **12**, **13**, **19**, **20**, **35**).-The presence of the *ortho*- or *para*-substituted phenyl substituent in the “1” position of 1,2,3-triazoles in benzimidazole-triazole hybrids can increase their antimicrobial activity.-In the case of benzimidazoles substituted in the “1” position with triazoles, the presence of an aliphatic or aromatic radical substituent increases the antimicrobial activity of the hybrids.-The presence of the oxygen atom in the bridge that connects the benzimidazole and triazole rings is favorable to the antimicrobial activity of the hybrids (compounds **19**, **20**, **21**, **29**, **30**).-The presence of the sulfur atom in the bridge that connects the benzimidazole and triazole rings is favorable to the antimicrobial activity of the hybrids and even to the antitubercular activity (**95**–**97**, **105**, **107**).-The presence of a supplementary triazole ring in benzimidazole-triazole hybrids improves their antimicrobial activity (compounds **43**, **45**, **47**).-The presence of the benzoyl substituent in the “5” position of the benzimidazole in the benzimidazole-1,2,4-triazole hybrids clearly improves their antimicrobial activity (compounds **85a**–**85e**).-The phenyl nucleus as a spacer between the “1” position of 1,2,4-triazole and the “2” position of benzimidazole favors the formation of antimicrobial compounds, and the substituents in the “5” position of the benzimidazole nucleus increase the antimicrobial activity (compounds **79**, **111**, **112**, **113**).-Only benzimidazole-1,2,3-triazole hybrids are mentioned in the literature as having antiviral properties.-2-Substituted or 1,2-disubstituted benzimidazoles with 1,2,3-triazoles are mentioned as antiviral compounds, and the presence of an additional triazole ring improves the antiviral activity (compound **140**).

The presence of both the benzimidazole ring and the triazole ring in a single molecule enhanced the effectiveness of the antimicrobial activities, as seen in the sections above. The recent ADME and SAR studies mentioned in this review are also important for directing new syntheses of benzimidazole-triazole hybrids in close correlation with their properties.

As mentioned in the cited literature, it is extremely useful, both from a therapeutic and economic point of view, that the synthesized compounds, such as the benzimidazole-triazole hybrids analyzed in this review, possess both antimicrobial and antimicrobial biological activity antiviral, to meet the medical requirements demanded especially lately, for better action, especially in the case of SARS-CoV-2.

The ADME studies performed on the benzimidazole-triazole hybrids mentioned in this review recommend the compounds as antimicrobials and antivirals and open new horizons to create new compounds, following the conclusions found here, with improved biological properties.

The articles researched on this topic, although they report the general characteristics of these molecules (lipophilicity/hydrophilicity), in order to have the desired antimicrobial or antiviral properties, refer only to liquid formulations in the form in which the compounds were tested, and so far not no article is reported that formulates in the form of nanosystems, nanoparticles for better availability of the active substance. This remains an open research topic for future studies.

We hope that this review will be useful for the design and synthesis of new benzimidazole-triazole hybrids with antimicrobial and antiviral properties in the context of exacerbation of microbial and viral infections and resistance to treatments with drugs known on the market.

## Data Availability

Not applicable.

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
