# Peer review of "Benzimidazole-Triazole Hybrids as Antimicrobial and Antiviral Agents: A Systematic Review"

_antibiotics, 2023, doi:10.3390/antibiotics12071220_

Round 1

Reviewer 1 Report

The manuscript tried to review on Benzimidazole-triazole hybrids as antimicrobial and antiviral agents. The author has considered the not really relevant and updated literature evidence in this scientific area and present their results and analysis in unclear and meaningful way. A few comments for the author are given below:

1. Title

Not appropriate. What kind of review article of this article-literature reviews, systematic reviews, or meta-analyses? Need to mention it in the title.

2. Abstract

The background is too short. It should be explain on the main objective of the study. In the methods and results of the abstract, please mention the results that has mentioned in the methods. More information about the highly significant observations should be clarified in the results. The author should provide conclusion with focusing on the innovations related to be in a harmony with the title of the manuscript and the objectives of your study. It looks like it does not reach a satisfactory level for a review article. Major improvements need to be made by the author.

3. Introduction

- Generally, introduction section needs several enhancements and error checks. I think the language should be rechecked by an English expert for grammar and typos.

- The author should focus on the objectives of their study. Introduce why she has chosen to study this topic is important with recent references.

- the introduction section in this manuscript has been written non-comprehensively. Author should outline the current situation and also evaluate the current situation (advantages/ disadvantages) and identify the gap.

- In the last paragraph, please add a couple of sentences to highlight the “innovative” aspect of the paper (added value in this field?) …what has been examined for very first time/or in more depth in this research area by your work presented in this manuscript?

- The length of each paragraph needs to be harmonized.

4. Results

- The author has not explained in detail about the findings.

5. Method

- The author did not explain in detail about the method in the manuscript

6. Discussion

- Generally this section needs to be supplemented with more other literature review papers. I think the author should discuss the innovations and more clarifying why she study in this section in more details.

- This manuscript is a review article. Therefore every statement needs to be supported with the correct reference source. It is strictly necessary to use the latest reference sources. The author needs to update the reference sources used. Make sure it is up to date.

- I feel it does not reach a satisfactory level for a review article. Major improvements need to be made by the author.

7. Conclusion: More details about her observations by showing some of this valuable study significations point by point should be more clarified.

8. References

- The number of references is very insufficient for a review article.

- A number or references need some formatting changes. Please ensure that the author follow MDPI guidelines for presenting name of authors/abbreviation of journals/year of publication in bold/volume journals in italics etc.

 9. Copyright issue

I noticed that there some figures and tables in this manuscript. Are these figures and tables being original work by the author? If material from other publications is reproduced in this manuscript, author should provide proof that she has obtained the necessary copyright permission. The author needs to state the source/copy right from where she got the pictures and tables that has been used in the manuscript, which is at the bottom of the figure and at the top of the table.  

Extensive editing of English language required. 

Author Response

The manuscript tried to review on Benzimidazole-triazole hybrids as antimicrobial and antiviral agents. The author has considered the not really relevant and updated literature evidence in this scientific area and present their results and analysis in unclear and meaningful way. A few comments for the author are given below: 

  1. Title

Not appropriate. What kind of review article of this article-literature reviews, systematic reviews, or meta-analyses? Need to mention it in the title.

 Response: Thank you for the remark. The article proposed to be a systematic review. That's why I added it in the title.

Benzimidazole-triazole hybrids as antimicrobial and antiviral agents:  A systematic review 

  1. Abstract

The background is too short. It should be explain on the main objective of the study. In the methods and results of the abstract, please mention the results that has mentioned in the methods. More information about the highly significant observations should be clarified in the results. The author should provide conclusion with focusing on the innovations related to be in a harmony with the title of the manuscript and the objectives of your study. It looks like it does not reach a satisfactory level for a review article. Major improvements need to be made by the author.

 Response: Thank you for the appreciation. The main objective of the study is included in the phrase:

"This review aims to provide an update on the synthesis methods of these hybrids, along with their antimicrobial and antiviral activities, as well as the structure–activity relationship reported in literature."

To complete the Abstract I added:

It was found that the presence of certain groups grafted onto the benzimidazole and/or triazole nuclei (-F, -Cl, -Br, -CF3, -NO2, -CN, -CHO, -OH, OCH3, COOCH3), as well as the presence of some heterocycles (pyridine, pyrimidine, thiazole, indole, isoxazole, thiadiazole, coumarin) increases the antimicrobial activity of benzimidazole-triazole hybrids. Also, the presence of the oxygen or sulfur atom in the bridge connecting the benzimidazole and triazole rings, generally increases the antimicrobial activity of the hybrids. The literature mentions only benzimidazole-1,2,3-triazole hybrids with antiviral properties. Both for antimicrobial and antiviral hybrids, the presence of an additional triazole ring increases their biological activity, which is in agreement with the three-dimensional binding mode of compounds. This review summarizes the advances of benzimidazole triazole derivatives as potential antimicrobial and antiviral agents covering articles published from 2000 to 2023.

  1. Introduction

- Generally, introduction section needs several enhancements and error checks. I think the language should be rechecked by an English expert for grammar and typos.

Response: Thank you for your observation. I corrected and rewrote the texts, as appropriate.

- The author should focus on the objectives of their study. Introduce why she has chosen to study this topic is important with recent references.

Response: Thank you for your observation. Taking into account your appreciation, I wrote:

Why did I choose the study of benzimidazole-triazole compounds? Classical drugs containing benzimidazole and triazole rings recommend these heterocycles as essential in building new target compounds with antimicrobial, antiviral, antiparasitic, etc. properties (Fig. 1). In addition, the literature mentions a series of benzimidazole-triazole hybrids with remarkable antimicrobial properties, antiviral activities, including new anti-SARS-COV-2 agents [70-74], with particular importance in the context of the recent pandemic, which led to the study of synthesis methods, antimicrobial properties, structure-property relationships and their biological activities. Therefore, this review aims to provide an update on the synthesis methods of the benzimidazole-triazole hybrids, along with their antimicrobial and antiviral activities, as well as the structure–activity relationship and DFT studies reported in literature.

            The objective of this study is described from the beginning of the article to its end, and it does not refer to anything else. This objective refers to:

- the synthesis of benzimidazole-triazole compounds and

- their antimicrobial properties and

 -antiviral properties.

-Syntheses of compounds and their biological properties are described successively,

-with an emphasis on the structure-property correlation,

-as well as DFT studies, where it is mentioned in the literature.

- the introduction section in this manuscript has been written non-comprehensively. Author should outline the current situation and also evaluate the current situation (advantages/ disadvantages) and identify the gap.

Response: Thank you for your observation. Taking into account your appreciation, I wrote:

The advantages of the study of benzimidazole-triazole hybrid compounds refer to a wider range of antimicrobial activities, compared to simple precursor heterocycles, to their better minimum inhibitory concentrations compared to simple component heterocycles, as well as to the need to hire specialized personnel to carry out these researches. The main disadvantages are material, because the synthesis of some hybrid compounds requires high costs compared to simple heterocycles, as well as greater time consumption. Consequently, if the synthesized hybrids have increased biological properties compared to simple precursor heterocycles, the balance clearly tilts towards the advantage of the synthesis of hybrid compounds. However, the access to hybrid compounds will not be without both sides, advantages and disadvantages, which requires a careful prospecting of all the components involved in the production of hybrids.

- In the last paragraph, please add a couple of sentences to highlight the “innovative” aspect of the paper (added value in this field?) …what has been examined for very first time/or in more depth in this research area by your work presented in this manuscript?

Response: Thank you for your observation. Taking into account your appreciation, I wrote:

This article summarizes for the first time in the literature: various synthesis methods of benzimidazole-1,2,3-triazole hybrids as well as benzimidazole-1,2,4-triazoles, their antimicrobial and antiviral activities, as well as SAR studies and DFT performed on the mentioned hybrids. Where necessary, for compounds with superior biological activities, several examples from the literature were given, and the various studies performed on them (in vitro, in vivo, in silico, etc.) were mentioned. All of these aim at directing the syntheses of hybrid compounds with specific structures and superior antimicrobial and antiviral properties, taking into account the mentions reported in the literature up to now.

- The length of each paragraph needs to be harmonized.

 Response: Thank you for the appreciation. I tried to harmonize the paragraphs of the article.

  1. Results

- The author has not explained in detail about the findings.

Response: Thank you for the appreciation. In the corrected version, I have explained the findings in as much detail as possible, as mentioned in the referenced articles.

  1. Method

- The author did not explain in detail about the method in the manuscript

Response: Thank you for your observation. Taking into account your appreciation, I wrote:

The database search methodology used in this review was the use of keywords, which can be found in the title, such as benzimidazole, 1,2,3-triazole, click reaction, 1,2,3-triazole, benzimidazole-triazole hybrids, antimicrobial, antiviral, or therapeutic properties, in different websites, such as PubMed, MDPI, Science Direct, Springer, The Royal Society Chemistry, ACS Publications, and Taylor & Francis. The selection of scientific articles for the last ten years was made according to the novelty brought in the benzimidazole-triazole hybids and their antimicrobial and antiviral properties, as well as the therapeutic properties of the reported compounds. In general, articles from the last ten years have been selected. For the hybrids found, first the syntheses and then their biological properties were presented, with special emphasis on those with improved properties (active on a larger range of microbial strains, with better minimum inhibitory concentrations, or where SAR studies were performed, DFT, etc).

  1. Discussion

- Generally this section needs to be supplemented with more other literature review papers. I think the author should discuss the innovations and more clarifying why she study in this section in more details.

Answer: Thank you for your observation. I completed the work with more data from the literature, with 54 reference articles in the field (185 References), research and reviews, and I discussed as detailed as possible about the innovations reported, of interest in the present review.

- This manuscript is a review article. Therefore every statement needs to be supported with the correct reference source. It is strictly necessary to use the latest reference sources. The author needs to update the reference sources used. Make sure it is up to date.

Answer: Thank you for your observation. All statements in the article are supported by the correct literature references found. I have used the most recent sources found. Thus, the article contains:

-36 articles from 2023

-24 articles from 2022

-22 articles from 2021

-18 articles from 2020

-16 articles from 2019

I searched and found the latest possible sources in the searched databases.

- I feel it does not reach a satisfactory level for a review article. Major improvements need to be made by the author.

Answer: Thank you for your observation. In addition to the previous version,

-I reviewed from all literature sources in the researched field, benzimidazole-triazole hybrids with antibacterial and antiviral properties, from which I mentioned the ADME, DFT and SAR studies performed and attached Figures. (with the necessary authorizations), of particular importance in the construction of new hybrids with biological properties.

-I highlighted from each source of literature the compounds with special biological activities and, in addition

-I noted these compounds with the best corresponding values, for example CMI on the Diagrams or Figures made. All Schemes and Figures, (with the exception of ADME or DFT in the articles), are ORIGINAL, being written in the ChemDraw 11 program.

- We carried out all the necessary discussions that refer to the variations of biological activities depending on the structures of the compounds, respectively, --------- the presence or absence of certain substituents on the studied molecules, in certain positions

- the presence or absence of certain additional heterocycles, connecting atoms in the studied molecules

- other data mentioned in the literature.

  1. Conclusion: More details about her observations by showing some of this valuable study significations point by point should be more clarified.

 Answer: Thank you for your observation. 

I have improved and enriched the conclusions of the review with new information regarding the data reported here, both

-in particular, on the structures of certain compounds, and

-in general, which refers to a wider range of data.

All changes are marked in the text.

  1. References

- The number of references is very insufficient for a review article.

Response: Thank you for your observation.

I researched several Reviews on Antibiotics, of the latest publications, and found the following:

  1. Article:

Patil, S.A.; Nesaragi, A.R.; Rodríguez-Berrios, R.R.; Hampton, S.M.; Bugarin, A.; Patil, S.A. Coumarin Triazoles as Potential Antimicrobial Agents. Antibiotics 2023, 12, 160. https://doi.org/10.3390/antibiotics12010160, has 66 references.

  1. Article:

Mariani, F.; Galvan, E.M. Staphylococcus aureus in Polymicrobial Skin and Soft Tissue Infections: Impact of Inter-Species Interactions in Disease Outcome. Antibiotics 2023, 12, 1164. https://doi.org/10.3390/antibiotics12071164, has 131 references.

  1. Article:

Syed, R.U.; Moni, S.S.; Break, M.K.B.; Khojali, W.M.A.; Jafar, M.; Alshammari, M.D.; Abdelsalam, K.; Taymour, S.; Alreshidi, K.S.M.; Elhassan Taha, M.M.; et al. Broccoli: A Multi-Faceted Vegetable for Health: An In-Depth Review of Its Nutritional Attributes, Antimicrobial Abilities, and Anti-inflammatory Properties. Antibiotics 2023, 12, 1157. https://doi.org/10.3390/antibiotics12071157, has 74 references.

  1. Article:

Miranda, C.; Igrejas, G.; Poeta, P. Bovine Colostrum: Human and Animal Health Benefits or Route Transmission of Antibiotic Resistance—One Health Perspective. Antibiotics 2023, 12, 1156. https://doi.org/10.3390/antibiotics12071156, has 63 references.

  1. Article:

Tao, R.E.; Prajapati, S.; Pixley, J.N.; Grada, A.; Feldman, S.R. Oral Tetracycline-Class Drugs in Dermatology: Impact of Food Intake on Absorption and Efficacy. Antibiotics 2023, 12, 1152. https://doi.org/10.3390/antibiotics12071152, has 42 references.

  1. Article:

Liu, C.; Monaghan, T.; Yadegar, A.; Louie, T.; Kao, D. Insights into the Evolving Epidemiology of Clostridioides difficile Infection and Treatment: A Global Perspective. Antibiotics 202312, 1141. https://doi.org/10.3390/antibiotics12071141 has 120 references.

  1. Article:

Severino, A.; Varca, S.; Airola, C.; Mezza, T.; Gasbarrini, A.; Franceschi, F.; Candelli, M.; Nista, E.C. Antibiotic Utilization in Acute Pancreatitis: A Narrative Review. Antibiotics 202312, 1120. https://doi.org/10.3390/antibiotics12071120 has 69 references

  1. 8. Article:

Karageorgos, S.; Hibberd, O.; Mullally, P.J.W.; Segura-Retana, R.; Soyer, S.; Hall, D., on behalf of the Don’t Forget the Bubbles. Antibiotic Use for Common Infections in Pediatric Emergency Departments: A Narrative Review. Antibiotics 202312, 1092. https://doi.org/10.3390/antibiotics12071092 has 129 references.

  1. 9. Article:

Santamaría-Corral, G.; Senhaji-Kacha, A.; Broncano-Lavado, A.; Esteban, J.; García-Quintanilla, M. Bacteriophage–Antibiotic Combination Therapy against Pseudomonas aeruginosaAntibiotics 202312, 1089. https://doi.org/10.3390/antibiotics12071089 has 60 references.

  1. Article:

Buzás, G.M.; Birinyi, P. Newer, Older, and Alternative Agents for the Eradication of Helicobacter pylori Infection: A Narrative Review. Antibiotics 202312, 946. https://doi.org/10.3390/antibiotics12060946 has 110 references.

These are among the latest reviews published in Antibiotics, and the list could go on...

However, considering the recommendation made, I have added a series of references of major importance for the research of Benzimidazole-triazole hybrids carried out in this review. The added references (54), along with all the others that have undergone changes, are marked on the text.

- A number or references need some formatting changes. Please ensure that the author follow MDPI guidelines for presenting name of authors/abbreviation of journals/year of publication in bold/volume journals in italics etc.

Response: Thank you for your observation.

 I have checked each citation separately, if it is written in accordance with MDPI requirements. I corrected the mistakes

  1. Copyright issue

I noticed that there some figures and tables in this manuscript. Are these figures and tables being original work by the author? If material from other publications is reproduced in this manuscript, author should provide proof that she has obtained the necessary copyright permission. The author needs to state the source/copy right from where she got the pictures and tables that has been used in the manuscript, which is at the bottom of the figure and at the top of the table. 

Response: Thank you for your observation.

-All 30 schemes are original.

-All 14 Tables are original.

-Among the figures that have been reproduced from the original, which are noted in the article, the necessary permissions are taken.

-The tables are made by me in Word.

-The schemes are made by me in ChemDraw.

-The figures are also made in ChemDraw, with the exception of those that are taken with permission from the articles.

Comments on the Quality of English Language

Extensive editing of English language required. 

Response: Thank you for your observation.

I have corrected the English Language.

Reviewer 2 Report

The current manuscript is an interesting and extensive review on the synthesis and use of benzimidazole-triazole hybrids as antimicrobial and antiviral agents. It is well presented and complete, especially in a chemical point-of-view. Nevertheless, some small changes should be made before acceptance for publication:

- I would like to see some images adapted from the cited articles (with permission rights granted from the original articles, of course), especially in what concerns antimicrobial and antiviral study results;

- The role of in silico approaches for the development of these types of molecules should be properly discussed (types of approaches, advantages/disadvantages, etc.);

- Formulation aspects and administration route aspects should also be discussed: given the general characteristics of these molecules (lipophilicity/hydrophilicity, vulnerability do metabolism, etc.), what types of formulations would be best – solid forms? Liquid ou semisolid forms? Nanosystems? And why.

Author Response

- I would like to see some images adapted from the cited articles (with permission rights granted from the original articles, of course), especially in what concerns antimicrobial and antiviral study results;

Response: Thank you for the remark.

I have attached some images adapted from the quoted articles (with permission granted from the original articles), regarding the treated subject, antimicrobial and antiviral studies.

- The role of in silico approaches for the development of these types of molecules should be properly discussed (types of approaches, advantages/disadvantages, etc.);

Response: Thank you for the remark.

For all the cases discussed in the literature, we realized the role of in silico approaches, with the necessary discussions on these types of molecules, as well as the approaches found, with the advantages and disadvantages encountered.

- Formulation aspects and administration route aspects should also be discussed: given the general characteristics of these molecules (lipophilicity/hydrophilicity, vulnerability do metabolism, etc.), what types of formulations would be best – solid forms? Liquid ou semisolid forms? Nanosystems? And why.

Response: Thank you for the remark.

The articles researched on this topic, although they report the general characteristics of these molecules (lipophilicity/hydrophilicity), in order to have the desired antimicrobial or antiviral properties, refer only to liquid formulations, in the form in which the compounds were tested, and so far not no article is reported that formulates in the form of nanosystems, nanoparticles for a better availability of the active substance. This remains an open research topic for future studies.

I added this observation in the Conclusions of the article.

Round 2

Reviewer 1 Report

I recommend that it be published without changes.